# Reversible multicolor chromism in layered formamidinium metal halide perovskites

Bryan A. Rosales [1], Laura E. Mundt[2], Taylor G. Allen [1], David T. Moore [1], Kevin J. Prince [1,3], Colin A. Wolden[3,4], Garry Rumbles[1,5], Laura T. Schelhas [2] & Lance M. Wheeler [1]✉

Metal halide perovskites feature crystalline-like electronic band structures and liquid-like physical properties. The crystal–liquid duality enables optoelectronic devices with unprecedented performance and a unique opportunity to chemically manipulate the structure with low energy input. In this work, we leverage the low formation energy of metal halide perovskites to demonstrate multicolor reversible chromism. We synthesized layered Ruddlesden-Popper $FA_{n+1}Pb_nX_{3n+1}$ (FA = formamidinium, X = I, Br; $n$ = number of layers = 1, 2, 3 … ∞) and reversibly tune the dimensionality ($n$) by modulating the strength and number of H-bonds in the system. H-bonding was controlled by exposure to solvent vapor (solvatochromism) or temperature change (thermochromism), which shuttles FAX salt pairs between the $FA_{n+1}Pb_nX_{3n+1}$ domains and adjacent FAX "reservoir" domains. Unlike traditional chromic materials that only offer a single-color transition, $FA_{n+1}Pb_nX_{3n+1}$ films reversibly switch between multiple colors including yellow, orange, red, brown, and white/colorless. Each colored phase exhibits distinct optoelectronic properties characteristic of 2D superlattice materials with tunable quantum well thickness.

[1] Center for Chemistry and Nanoscience, National Renewable Energy Laboratory, 15013 Denver West Parkway, Golden, CO 80401, USA. [2] SLAC National Accelerator Laboratory, 2575 Sand Hill Road, Menlo Park, CA 94025, USA. [3] Department of Chemical and Biological Engineering, Colorado School of Mines, Golden, CO 80401, USA. [4] Material Science Program, Colorado School of Mines, Golden, CO 80401, USA. [5] Renewable and Sustainable Energy Institute, Department of Chemistry, University of Colorado Boulder, Boulder, CO 80309, USA. ✉email: lance.wheeler@nrel.gov

Metal halide perovskite (MHP) materials have captured the imagination of the scientific community[1–5]. The coexistence of crystalline-like electronic band structure and liquid-like physical properties of MHP materials endows them with long carrier lifetimes, long carrier diffusion lengths, and exceptional defect tolerance to enable an array of exciting optoelectronic applications[6]. However, it is challenging to prevent changes in phase or composition of these "crystalline liquids"[7] even in benign conditions. Despite issues with stability upon exposure to humidity, heat, and light[8–10], MHPs have enabled solar cells with a record power conversion efficiency (PCE) > 25%[11].

Three-dimensional (3D) MHPs are composed of corner-sharing $[MX_6]^{4-}$ octahedra ($M = Pb^{2+}, Sn^{2+}; X = I^-, Br^-, Cl^-$) that form an isotropic inorganic framework separated by A-site organic (methylammonium (MA), formamidinium (FA), etc.) or inorganic (alkali metal) monovalent cations. 2D MHPs also exhibit corner-sharing $[MX_6]^{4-}$ octahedra but only in two dimensions with the third dimension separated in space by the A-site cation[12–14]. 2D MHPs are classified as Ruddlesden-Popper[15] or Dion-Jacobson[16,17] phases of the general formula $A'_2A_{n-1}M_nX_{3n+1}$ or $A'A_{n-1}M_nX_{3n+1}$, respectively, where $A' = 1+$ or $2+$ cation, $A = 1+$ cation, and $n = 1$, 2, 3, … $\infty$ is the number of connected 2D octahedra layers.

Chromogenic compounds change visible color when subjected to an external stimulus such as light, temperature, electric potential, solvent/vapor, or mechanical forces[18–20]. Materials that chromogenically change under these stimuli are classified as photochromic, thermochromic, electrochromic, solvatochromic, and mechanochromic, respectively[18]. Realizing reversible chromism in MHPs unlocks a new class of functional materials that couples a dynamic element to their remarkable optoelectronic properties. We envision dynamically tunable semiconductors to have applications that span switchable photovoltaics[21,22] to energy storage[23] and neuromorphic computing[24]. To date, there are two recognized mechanisms for reversible chromism in MHP materials: (1) crystal phase transformation[22] and (2) molecular intercalation[21]. Both of these mechanisms have enabled the first examples of switchable photovoltaic windows[21,22], which circumvent the fundamental trade-off between power conversion and visible light transmittance of traditional photovoltaic windows[25].

MHPs are inherently thermochromic materials exhibiting significant optical changes induced by crystal phase transitions between the α-phase, a black high-symmetry perovskite phase composed of corner-sharing $[MX_6]^{4-}$ octahedra, and the δ-phase, a yellow-to-colorless non-perovskite hexagonal or orthorhombic phase composed of face- or edge-sharing $[MX_6]^{4-}$ octahedra, respectively[26–28]. Utilizing this phase transformation, the mixed-halide perovskite $CsPbI_{0.5}Br_{2.5}$ was employed in switchable photovoltaic window that achieved a 4.69% PCE with a phase transition temperature of 105 °C[22]. However, it is challenging to achieve transition temperatures below 100 °C for thermochromic window applications, and the smaller thermodynamic driving force at low temperature results in many hours for a complete phase transition[22]. In contrast, exposure of MHPs to vapor molecules that interact with the lattice can greatly decrease the temperature necessary for chromogenic behavior and improves transition speeds to seconds or minutes instead of hours as a result of the low formation/dissociation energy inherent in MHPs[29–31]. We have shown previously that exposure of $MAPbI_3$ to methylamine vapor generates the clear $MAPbI_3 \cdot xCH_3NH_2$ complex, which can be switched back to black $MAPbI_3$ when heated with solar illumination[21]. Others have shown that room temperature solvatochromism is possible through the intercalation of ethanol into 2D $OA_2MAPb_2I_7$[32] or by forming reversible hydrates for the transition between 3D $MAPbBr_3$ and 0D $MA_4PbBr_6 \cdot 2H_2O$[33]. FA-based MHPs, unlike MA analogues, do not exhibit hydrate phases. Instead, exposure of α-$FAPbI_3$ to $H_2O$

catalytically converts the film into δ-$FAPbI_3$ by interacting with the anisotropically strained (111) α-$FAPbI_3$ lattice plane[34–36].

In this work, we synthesize composite films composed of layered FA-based MHPs of the general formula $FA_{n+1}Pb_nX_{3n+1}$ ($X = I, Br$) and their mixed-halide compositions. We leverage their liquid-like physical properties to demonstrate a third mechanism of dynamic chromism in MHPs through reversible layer formation and coalescence to form compounds that span 2D $FA_2PbX_4$ ($n = 1$) to 3D α-$FAPbX_3$ ($n = \infty$) and finally 1D δ-$FAPbI_3$. Reversible chromism is enabled by a second "reservoir" phase in the film composed of excess FAX salt, which allows FAX to reversibly shuttle between the reservoir and MHP layers. The mechanism is controlled by modulating the strength and number of H-bonds between the reservoir, MHP, and solvent vapor. Unlike previous reports on switchable MHPs that only switch between a single dark and a single light color[21,22], our composite $FA_{n+1}Pb_nX_{3n+1}$ films reversibly switch between a continuum of colors spanning yellow, orange, red, brown, and white/colorless. The colored phases are 2D superlattice materials with tunable quantum well thickness.

## Results

**Synthesis of layered formamidinium metal halide perovskites.** Synthesis methods for free-standing powder and thin films of layered formamidinium metal halide perovskites were developed. We synthesized $FA_{n+1}Pb_nX_{3n+1}$ powder by ball-milling a mixture of 4:1 FAX:$PbX_2$ for 60 min in an inert atmosphere. Wide-angle X-ray scattering (WAXS) data shows that $FA_{n+1}Pb_nI_{3n+1}$ powder exhibits Bragg diffraction peaks that correspond to a mixture of 2D $FA_2PbI_4$ ($n = 1$) with staggered octahedral layers and 3D α-$FAPbI_3$ (3C/3R) that corresponds to $n \geq 2$ (Supplementary Fig. 1). Though predicted to exist[37], we believe this is the first report of iodide compositions of $FA_{n+1}Pb_nX_{3n+1}$, and our WAXS results are in excellent agreement with previously reported $MA_2PbI_4$[38] and $FA_2PbBr_4$[39]. Several different structures have been observed during the formation of FA-based halide perovskites, including 2H, 4H, 6H, 3C, and 3R structures (Supplementary Fig. 2)[40]. Our WAXS data eliminates any significant contributions from hexagonal structures (2H, 4H, 6H). In the $FA_{n+1}Pb_nX_{3n+1}$ compounds observed here, $FA^+$ molecules separate the 2D layers rather than the long-chain or bulky cations typically used in other Ruddlesden-Popper phases, such as butylammonium (13.4 Å), phenylethylammonium (16.6 Å), and hexylammonium (18.4 Å)[41,42]. Here we observe a smaller interlayer spacing of 8.9 Å, determined from the (001) peak of $FA_2PbI_4$, which is approximately the length of two $FA^+$ molecules (8.2 Å, see Methods section).

We produce functional thin films and expand to bromide alloys by developing a scaffold composite composed of $FA_{n+1}Pb_nX_{3n+1}$, an FAX reservoir, and a porous $Al_2O_3$ nanoparticle scaffold (Fig. 1a). Composite $FA_{n+1}Pb_nX_{3n+1}$ films are synthesized by spin-coating a precursor solution containing 3 M FAX and 0.75 M $PbX_2$ (4:1 FAX:$PbX_2$; $X = I, Br$) in DMSO onto a 1.58 ± 0.02 μm-thick $Al_2O_3$ scaffold. $FA_{n+1}Pb_nX_{3n+1}$ domains are formed by annealing the spin-coated film at 60 °C for 10 min. WAXS of composite $FA_{n+1}Pb_nX_{3n+1}$ films (Fig. 1b) match $FA_{n+1}Pb_nI_{3n+1}$ powder produced by ball-milling (Supplementary Fig. 1) and both exhibit a mixture of Bragg diffraction peaks that correspond to 2D $FA_2PbX_4$ ($n = 1$) with staggered octahedral layers and 3D α-$FAPbX_3$ (3C/3R). Specifically, we observe Bragg diffraction peaks at 0.703 Å$^{-1}$, 1.400 Å$^{-1}$, 2.040 Å$^{-1}$, and 2.095 Å$^{-1}$ for 100% I and 0.738 Å$^{-1}$, 1.478 Å$^{-1}$, 2.157 Å$^{-1}$, and 2.198 Å$^{-1}$ for 100% Br that correspond to the (001), (002), (201), and (003) planes of 2D $FA_2PbX_4$ ($n = 1$) (Fig. 1b, asterisks). The 100% Br sample exhibits peaks shifted to higher Q due to the smaller lattice constant associated the bromide anion.

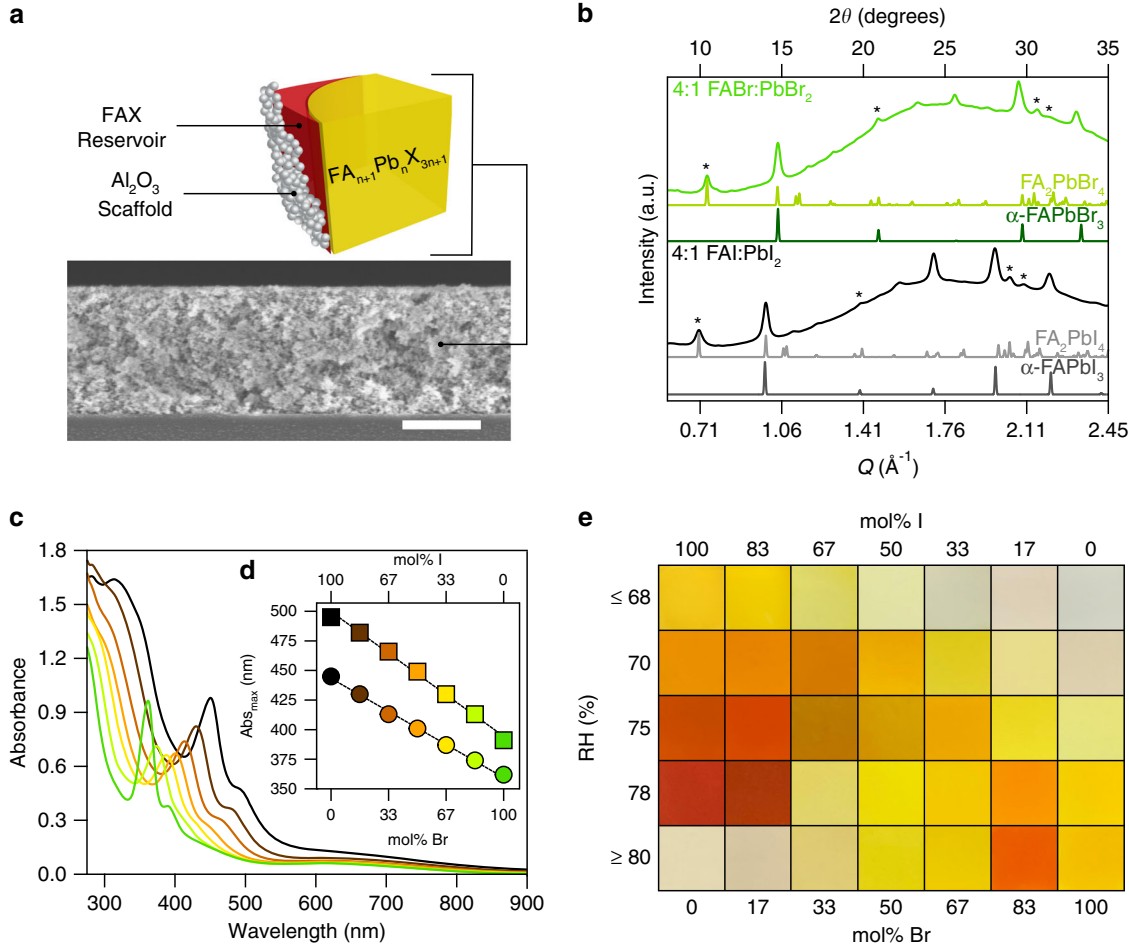

**Fig. 1 FA$_{n+1}$Pb$_n$X$_{3n+1}$ composite film characterization and reversible chromism. a** Illustration showing the components of a FA$_{n+1}$Pb$_n$X$_{3n+1}$ composite film, including FA$_{n+1}$Pb$_n$X$_{3n+1}$, FAX reservoir, and Al$_2$O$_3$ scaffold. Scale bar is 1 μm. **b** Wide angle X-ray scattering (WAXS) patterns of composite FA$_{n+1}$Pb$_n$X$_{3n+1}$ films. The 2$\theta$ axis in (**b**) is relative to Cu K$\alpha$ (1.5406 Å, 8.04 eV) radiation and was calculated from $Q = 4\pi\sin(\theta)/\lambda$ where $\lambda$ is the excitation wavelength. The background scattering in (**b**) is caused by the Al$_2$O$_3$ scaffold and glass substrate. * = FA$_2$PbX$_4$. Reference patterns of $\alpha$-FAPbI$_3$ were obtained from the Materials Design Group[78], and the reference pattern of FA$_2$PbI$_4$ was obtained from ref. [37]. Reference pattern of FA$_2$PbBr$_4$ was generated from FA$_2$PbI$_4$ by substituting Br in place of I and multiplying the unit cell volume by the Br/I radius ratio. **c** Absorbance spectra and **d** absorbance maxima as a function of composition for FA$_{n+1}$Pb$_n$X$_{3n+1}$ films. Circles in **d** correspond to the strong excitonic peak tunable between 450 and 360 nm ($n = 1$) while squares correspond to the absorption edge tunable between 500 and 390 nm ($n > 1$), respectively. **e** Representative optical photographs of composite FA$_{n+1}$Pb$_n$X$_{3n+1}$ films highlighting the diverse colors obtainable upon exposure to humidity. [PbI$_2$] = 0.75 M in dimethyl sulfoxide (DMSO) is held constant in all precursors, which were spun onto a 1.58 ± 0.02 μm Al$_2$O$_3$ scaffold and annealed at 60 °C for 10 min. Note that films rich in iodide (>80% I) or bromide (>67% Br) bleach to white/transparent upon exposure to >80% or <68% relative humidity (RH), respectively.

The optically active crystalline FA$_{n+1}$Pb$_n$X$_{3n+1}$ (X = I, Br) domains are composed of predominantly $n = 1$ and $n = 2$ layers. FA$_{n+1}$Pb$_n$X$_{3n+1}$ domains exhibit compositionally tunable optical properties that vary linearly as the halide ratio is varied from 100% iodide to 100% bromide (Fig. 1c, d) with two absorption peaks: (1) a strong excitonic peak tunable between 450 and 360 nm (circles) and (2) an absorption edge tunable between 500 and 390 nm (squares) that correspond to $n = 1$ and $n = 2$ layers, respectively. The optical properties observed for composite FA$_{n+1}$Pb$_n$X$_{3n+1}$ films are consistent with other 2D MHPs[41,43–46].

**Reversible chromism in layered FA-based MHPs.** Composite FA$_{n+1}$Pb$_n$X$_{3n+1}$ films show a brilliant array of reversible coloration upon exposure to solvent vapor. The visual appearance of composite FA$_{n+1}$Pb$_n$X$_{3n+1}$ films range from colorless to yellow, orange, red, and brown (Fig. 1e). We attribute the mechanism of color change to a dynamic equilibrium between FAX salt pairs intercalated into the FA$_{n+1}$Pb$_n$X$_{3n+1}$ domain or shuttled to an adjacent amorphous "reservoir" domain composed of excess FAX

(Fig. 1e). The equilibrium is described by Eq. 1 (see Supplementary Note 1 for a derivation of $q$):

$$FA_{n+1}Pb_nX_{3n+1} \leftrightarrow FA_{n+1-q}Pb_nX_{3n+1-q} + qFAX \quad (1)$$

$$n = 1, 2, 3, \ldots \infty; q = \frac{1}{n(n+1)}$$

The dynamic equilibrium is shifted by modulating the strength and number of H-bonds between solvent molecules, reservoir, and MHP. This insight allows us to design composite materials composed of FA$_{n+1}$Pb$_n$X$_{3n+1}$ domains that change color in response to solvent vapor (solvatochromism) or temperature (thermochromism) by reversibly forming compounds that span 2D FA$_2$PbX$_4$ ($n = 1$) to 3D $\alpha$-FAPbX$_3$ ($n = \infty$) and finally 1D $\delta$-FAPbI$_3$ (Fig. 2).

There are three main drivers in the design of composite thin films: (1) FA$_{n+1}$Pb$_n$X$_{3n+1}$ domains must be accessible to solvent vapor, (2) a reservoir phase must be present, and (3) the solvent vapor must interact favorably with FA$_{n+1}$Pb$_n$X$_{3n+1}$. In order to

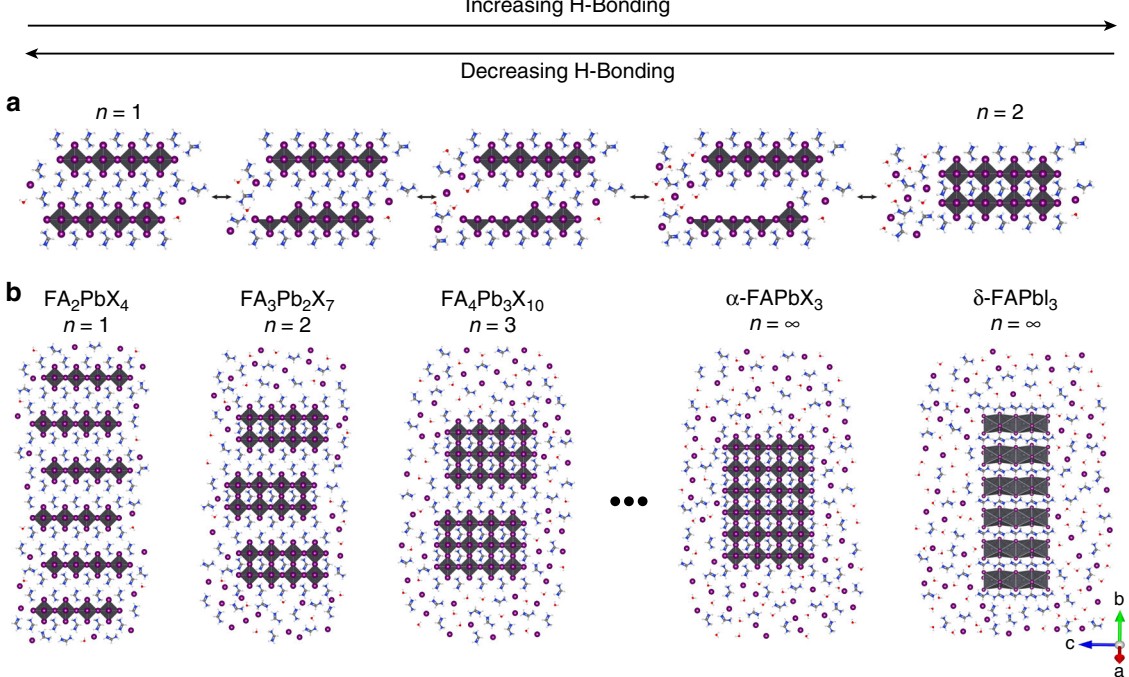

**Fig. 2 Mechanism of multicolor chromism in layered formamidinium metal halide perovskites. a** Reversible FAX shuttling between the FAX reservoir and layered metal halide perovskite (MHP) causes layer formation and coalescence. **b** Illustration of the reversible switching mechanism in chromic composite $FA_{n+1}Pb_nX_{3n+1}$ (X = I, Br) films enabled by a FAX reservoir adjacent to $FA_{n+1}Pb_nX_{3n+1}$ domains. Each colored film is a mixture of multiple thicknesses (n).

provide solvent access, $FA_{n+1}Pb_nX_{3n+1}$ domains are formed in a mesoporous $Al_2O_3$ nanoparticle scaffold. The scaffold limits the size of the $FA_{n+1}Pb_nX_{3n+1}$ domains and provides a pathway for solvent vapor transport. The reservoir phase is formed by using an excess of FAX relative to $PbX_2$ (>3:1 FAX: $PbI_2$). Solvent molecules and water in the composite film are maintained by annealing at low temperatures (60 °C) in a 40% relative humidity (RH) atmosphere without exposure to dry-air flow. A detailed discussion and explanation for composite $FA_{n+1}Pb_nX_{3n+1}$ film formation is supplied in Supplementary Note 2 and Supplementary Figs. 3–9. When assembled properly, composite films will reversibly change color rapidly (<1–20 min depending on solvent vapor flow rate) and over many cycles (>10). Though ball-milled powder samples show a chromogenic response to solvent vapor, the transitions are slower (30 min–1 h) with limited reversibility (Supplementary Fig. 10).

The final design criterium is choice of solvent vapor. Reversible chromism of composite $FA_{n+1}Pb_nX_{3n+1}$ films is strongly dependent on the affinity of the solvent vapor to form H-bonds with the FAX reservoir and $FA_{n+1}Pb_nX_{3n+1}$ constituents. A variety of solvent vapors were used to highlight the dependence on H-bonding (Supplementary Fig. 11). Chloroform and dichloromethane (DCM) have Lewis acidic protons that are H-bond donors but have no significant H-bond acceptor character. Tetrahydrofuran (THF) and pyridine have oxygen or nitrogen Lewis basic groups, respectively, that are H-bond acceptors. Composite $FA_{n+1}Pb_nX_{3n+1}$ films exposed to chloroform, DCM, or THF do not noticeably change color, whereas pyridine bleaches the film to colorless. The stronger Lewis basicity of the pyridine compared to THF leads to solvation and complex formation with $[PbX_6]^{4-}$ to form isolated octahedra. The ability of the solvent to both donate and accept H-bonds is critical for reversible switching to occur. Solvent vapors such as water or alcohols (methanol, ethanol, and isopropyl alcohol) with H-bond-active hydroxyl groups

will deeply color composite $FA_{n+1}Pb_nX_{3n+1}$ films from yellow to orange, red, and brown (Supplementary Fig. 11).

**Molecular interactions in reversible chromism**. H-bonds are the thermodynamic driving force for the reversible chromism in composite $FA_{n+1}Pb_nI_{3n+1}$ films. Chromism is dependent on both the strength and number of H-bonds. For instance, color change induced by exposure to solvent vapor (solvatochromism) may be reversed by increasing the temperature of the film (thermochromism) if the vapor concentration is held constant. The thermochromic effect is a function of the strength of H-bonding in the system, as the relative strength of H-bonds is weakened when heated and strengthened when cooled. We observe this effect with only mild conditions where a bleached film will rapidly (<10 s) convert to brown upon heating the film to 35 °C (Supplementary Fig. 12), which nicely resides in the desirable range for thermochromic window applications[21].

The remainder of the discussion centers around chromism induced by a change in the number of H-bonds, rather than strength, to manipulate the thermodynamic equilibrium of Eq. 1. We control the number of H-bonds by either changing the relative amount of the $FA_{n+1}Pb_nX_{3n+1}$ phase to the reservoir phase or by changing the concentration of solvent vapor in the environment. We chose $FA_{n+1}Pb_nI_{3n+1}$/water vapor as a model system since water vapor affords facile control over RH (concentration) compared to the partial pressure of other solvents and fast solvatochromic (hygrochromic) switching of <1–20 min depending on water vapor flow rate. A switching time of <7 s is shown in Supplementary Movie 1. We believe these observations are representative of the underlying mechanism, regardless of choice of chemical system.

We probe molecular interactions in the $FA_{n+1}Pb_nI_{3n+1}$/water vapor model system by varying the RH between 17 and 83% RH in conjunction with in-situ diffuse-reflectance infrared Fourier

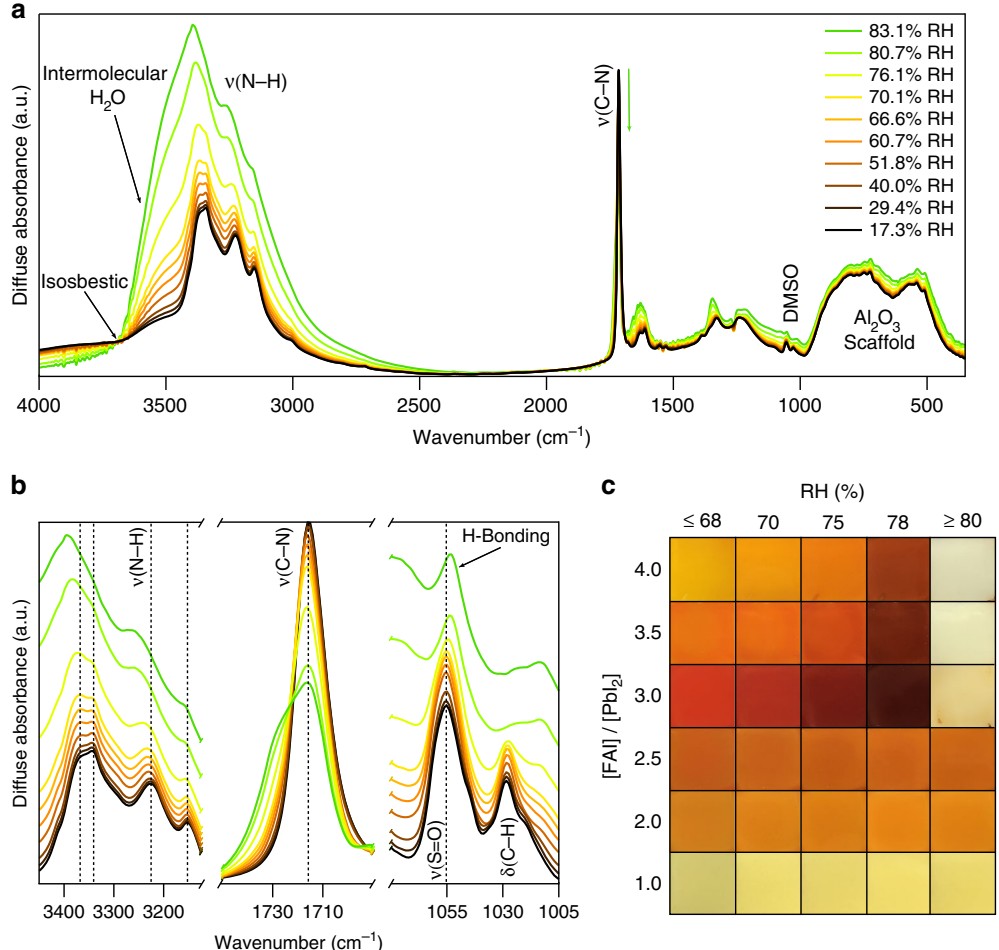

**Fig. 3 Molecular interactions are the thermodynamic driving force for reversible chromism. a** Representative diffuse-reflectance infrared Fourier transform spectroscopy (DRIFTS) spectra collected at various relative humidity (RH) and extracted from Supplementary Fig. 13. **b** Spectra focused on the H-bonding regions of the composite film: N–H stretch region of FA$^+$, C–N asymmetric stretch region of FA$^+$, and the S=O stretch and C–H rocking regions of dimethyl sulfoxide (DMSO). Dashed vertical lines indicate peak location at 17.3% RH. **c** Optical photographs showing the effect of changing the number of molecular interactions on the hygrochromic properties by varying the FAI:PbI$_2$ ratio and RH. [PbI$_2$] = 0.75 M in DMSO was held constant in all precursors, which were spun onto a 1.58 ± 0.02 μm Al$_2$O$_3$ scaffold and annealed at 60 °C for 10 min. DRIFTS spectra and optical photographs were collected on films with a FAI:PbI$_2$ ratio of 4.0.

transform spectroscopy (DRIFTS). The DRIFTS spectrum collected at 17.3% RH clearly shows vibrational modes corresponding to the components of the reservoir. The FAI is represented by N–H stretching modes between 3400 and 3200 cm$^{-1}$ and C–N asymmetric stretching at 1716 cm$^{-1}$ correspond to the FA$^+$ cation[47]. We also observe DMSO, H$_2$O, and the Al$_2$O$_3$ scaffold (Fig. 3a, black spectrum): C–H bending modes between 1400 and 1300 cm$^{-1}$, S=O stretching at 1056 cm$^{-1}$, and C–H rocking at 1028 cm$^{-1}$ corresponding to DMSO[48]; H–O–H bending at 1630 cm$^{-1}$ and O–H stretching at 3510 cm$^{-1}$ corresponding to intermolecularly bound H$_2$O[49,50]; and Al-O stretching between 1000 and 500 cm$^{-1}$ corresponding to the Al$_2$O$_3$ scaffold[51].

We increase the number of H-bonds by increasing RH from 17.3% to 83.1%. The film changes color from yellow to orange to brown to white/colorless (Fig. 3c, top row), and the DRIFTS spectra correspondingly undergo several changes. Before a change in color occurs (RH < 70%), the increased intensity of the O–H stretching mode located between 3550 and 3200 cm$^{-1}$ suggests increasing amounts of H$_2$O are incorporated into the film and are intermolecularly H-bonded with other H$_2$O molecules or with FAI in the reservoir[49,50]. The isosbestic point at 3688 cm$^{-1}$ signifies a reduction in scaffold-bonded H$_2$O (>3688 cm$^{-1}$) and

an increase in intermolecularly H-bonded H$_2$O or hydrated FAI (3550–3200 cm$^{-1}$, Fig. 3a)[48].

We observe a threshold value for chromism at c.a. 70% RH, which results in more dramatic changes to the DRIFTS spectra. The H-bond interactions shift the thermodynamic equilibrium (Eq. 1) from the FA$_{n+1}$Pb$_n$I$_{3n+1}$ domains to the FAI reservoir, indicated by an increase in spectral intensity, peak shifting, and peak broadening at RH > 70%. The N–H stretching modes broaden and blueshift, indicating the bonding environment of FA$^+$ is changing over time as it is shuttled from the FA$_{n+1}$Pb$_n$X$_{3n+1}$ domain to the reservoir (Fig. 3b). The broadening is accompanied by a shoulder that grows in at 1730 cm$^{-1}$ (Fig. 3b). The blueshift at high RH indicates both shortening and strengthening of the N–H bond[52] as a result of H-bond formation. Both S=O stretch and C–H rocking modes of DMSO remaining in the film redshift at higher RH values, which is characteristic with H-bonding to water and a distinct change in bonding environment (Fig. 3b)[48]. Although solvatochromism in our ball-milled powder synthesized without solvent (Supplementary Fig. 10) shows that DMSO is not crucial to the reversible switching mechanism, DMSO may facilitate switching by providing H-bond acceptor sites. The molecular interactions observed in DRIFTS during color change from yellow to orange, brown, and colorless are

reversed upon removal of water vapor (Supplementary Fig. 13). Removal of H-bond interactions shift the thermodynamic equilibrium (Eq. 1) from the reservoir back to the $FA_{n+1}Pb_nI_{3n+1}$ domains.

The reservoir volume also dictates the number of molecular interactions and the resulting color of the $FA_{n+1}Pb_nI_{3n+1}$ composite. Hygrochromic properties are only observed when the $FAI:PbI_2$ ratio is >3.0, which suggests that enough FAI must be present to form both the reservoir and the layered $FA_{n+1}Pb_nI_{3n+1}$. As the $FAI:PbI_2$ ratio increases, the number of molecular interaction sites increases in the film. The color of the film changes from red to orange to yellow as the equilibrium is pushed to thinner layers (smaller $n$) with higher $FAI:PbI_2$ ratios. Thus, by controlling the number of molecular interactions through varying both the RH and the $FAI:PbI_2$ ratio, the chromic response of the film can be extensively controlled with a wide variety of colors obtainable, including yellow, orange, red, brown, dark brown, and white/transparent. We note the 2D nature of our films lead to significantly improved moisture stability as evidenced by repeated and reversible color cycling between 20 and 82% RH, storage at

≤40% RH in air for months, and storage at 75% RH in air for over one month.

**Structural evolution due to molecular interactions.** We probe the structural response of $FA_{n+1}Pb_nI_{3n+1}$ composites induced by molecular interactions using in-situ WAXS experiments. In-situ WAXS experiments nicely illustrate the reversible transformation from predominantly $FA_{n+1}Pb_nI_{3n+1}$ domains with $n = 1, 2$ to α-FAPbI$_3$ ($n = \infty$) and finally δ-FAPbI$_3$. Starting at ambient RH, the films are exposed to humid air controlled at 82% RH. Dry conditions (~40% RH) are achieved by flowing Helium. The films are cycled through humid/dry conditions three times during collection of in-situ WAXS data (Fig. 4a). Flowing humid air at 82% RH causes Bragg peaks corresponding to $FA_2PbI_4$ and α-FAPbI$_3$ to gradually disappear over 10.5 min and 18 min, respectively, while Bragg peaks corresponding to δ-FAPbI$_3$ begin to emerge at 9 min. Disappearance of $FA_2PbI_4$ peaks in the WAXS pattern before α-FAPbI$_3$ peaks disappear suggests $n = 1$ layers coalesce to form layers of $n > 1$.

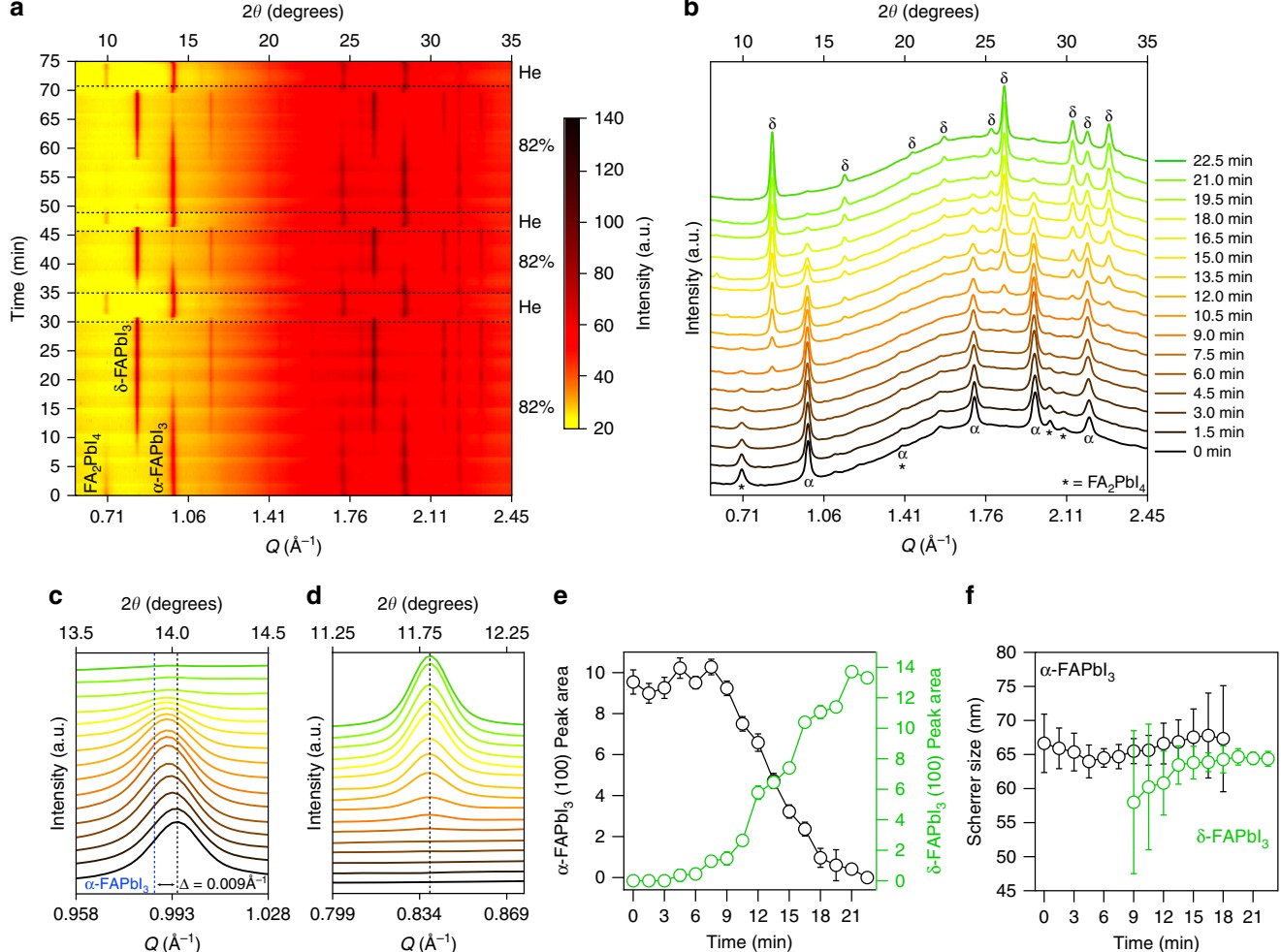

**Fig. 4 Structural evolution during reversible hygrochromic cycling. a** In-situ wide-angle X-ray scattering (WAXS) data collected on a $FA_{n+1}Pb_nI_{3n+1}$ composite film over three cycles of alternating exposure to 82% relative humidity (RH) and He flow. **b** Select WAXS data extracted from (**a**). Zoom-in on the region corresponding to the (100) peak of cubic α-FAPbI$_3$ (**c**) and to the (100) peak of hexagonal δ-FAPbI$_3$ (**d**). The 2θ axes are relative to Cu Kα (1.5406 Å, 8.04 eV) radiation and was calculated from $Q = 4\pi\sin(\theta)/\lambda$ where $\lambda$ is the excitation wavelength. The WAXS background scattering is caused by the $Al_2O_3$ scaffold and glass substrate, which is necessary for chromic properties as discussed in Supplementary Note 2. Comparison of cubic α-FAPbI$_3$ (100) and hexagonal δ-FAPbI$_3$ (100) peak area (**e**) and Scherrer crystallite size (**f**) as a function of time exposed to 82% RH. Fits and s.d. error bars in (**e**, **f**) were obtained by fitting the cubic α-FAPbI$_3$ or hexagonal δ-FAPbI$_3$ (100) peaks in (**b**) to a Voigt function.

Surface energy and strain are known to stabilize small or 2D grains of α-FAPbI$_3$, whereas bulk crystals thermodynamically favor the non-perovskite δ-FAPbI$_3$ hexagonal phases[34,53]. The strain of the domains is reduced as the domain grows and eventually transforms into the thermodynamically favored δ-FAPbI$_3$ phase. We observe a strain-induced shift in the (100) peak of α-FAPbI$_3$ from 0.995 Å$^{-1}$ to 0.991 Å$^{-1}$ ($\Delta = 0.004$ Å$^{-1}$) as RH is increased due to a relaxation of the growing α-FAPbI$_3$ crystal between FAI layers (Fig. 4c)[34]. In contrast to the (100) peak of α-FAPbI$_3$ at 0.995 Å$^{-1}$, the (100) peak of δ-FAPbI$_3$ at 0.838 Å$^{-1}$ does not exhibit a shift in $Q$ due to the relatively nonexistent strain in the 1D δ-FAPbI$_3$ lattice (Fig. 4d)[34]. We note that the δ-FAPbBr$_3$ phase is not observed at high RH (Supplementary Fig. 14) because δ-FAPbBr$_3$ is only stable below −8 °C (265 K)[54].

The WAXS data clearly shows hygrochromism in composite FA$_{n+1}$Pb$_n$I$_{3n+1}$ films is not due to recrystallization of the material (dissolution of one phase followed by precipitation of another), but rather by reversible FAI shuttling between grains that maintain their initial size. We tracked the evolution of the (100) peak area for each species as a function of time exposed 82% RH (Fig. 4e) and quantified the size of the FA$_{n+1}$Pb$_n$I$_{3n+1}$ domains using Scherrer analysis (Fig. 4f)[55]. The α-FAPbI$_3$ (100) peak area decreases while the δ-FAPbI$_3$ (100) peak area increases with an intersection occurring between 12-15 min, indicating the transformation of a single grain rather than the formation of a new one. 2D WAXS data collected during phase evolution shows the grains are polycrystalline and do not become textured over time (Supplementary Fig. 15), and the size of the domains remain constant as the FA$_{n+1}$Pb$_n$I$_{3n+1}$ domains evolve with increasing RH (Fig. 4f). After exposure to flowing humid air at 82% RH until the film is white/colorless, exposure of flowing dry He reverses the observed phase transformations: the (100) peak of δ-FAPbI$_3$ at 0.838 Å$^{-1}$ disappears while the (100) peak of α-FAPbI$_3$ at 0.995 Å$^{-1}$, and the (001), (002), (201), and (003) peaks of FA$_2$PbI$_4$ at 0.703 Å$^{-1}$, 1.400 Å$^{-1}$, 2.040 Å$^{-1}$, and 2.095 Å$^{-1}$ reemerge (Fig. 4a). As we remove H$_2$O from the film, FAI is re-inserted into the α/δ-FAPbI$_3$ domains to re-form the layered FA$_{n+1}$Pb$_n$I$_{3n+1}$ structure.

**Dynamic control of optoelectronic properties.** Optical spectroscopy is used to probe the optical properties of the FA$_{n+1}$Pb$_n$I$_{3n+1}$ domains as the $n = 1$ material coalesces to higher dimensional structures ($n > 1$). Our optical absorption and photoluminescence (PL) data confirm each observed color is a FA$_{n+1}$Pb$_n$I$_{3n+1}$ mixture with multiple thicknesses that span $n = 1$ to $n = \infty$ (Fig. 5a). Varying the RH changes the relative ratios of n as the FA$_{n+1}$Pb$_n$I$_{3n+1}$

domains trend toward thicker layers with increasing RH. Increasing the RH up to 78% results in a decrease of the excitonic peak at 2.79 eV and an increase in the absorption edge at 2.51 eV as well as the emergence of new absorption edges located at 2.30 eV, 2.07 eV, and 1.77 eV (Fig. 5a). Subjecting the film to 80% RH causes a decrease in all peaks as the film transforms to white/colorless. A dramatic absorption change is observed upon reaching 82% RH in which absorbance in the visible region is greatly reduced and a strong absorbance peak located at 3.18 eV emerges that is consistent with formation of hexagonal δ-FAPbI$_3$[28,56]. This large distribution of absorption properties obtainable by varying RH is reflected in the CIE 1976 L*a*b* coordinate space that is representative of the human perception of color (Fig. 5b).

Discrete optical transitions observed in the absorbance spectra occur due to the separation or coalescence of 2D octahedra layers. The optical bandgap of 2D FA$_{n+1}$Pb$_n$I$_{3n+1}$ materials increases monotonically as $n$ approaches 1 due to formation of minibands in the quantum well superlattice structure that emerges from alternating layers of formamidinium and connected lead halide octahedra layers (Fig. 6a). The optical bandgap increase relative to 3D α-FAPbI$_3$ is written as:

$$E_{g,2D} = E_{g,3D} + E_e + E_h \tag{2}$$

where $E_{g,3D} = 1.52$ eV is the bulk bandgap of α-FAPbI$_3$[57] and $E_{e(h)}$ is the minimum energy of the lowest-energy miniband. We determine $E_{e(h)}$ by adapting the Kronig–Penney (KP) model[58] for an electron (hole) in a one-dimensional periodic potential. The KP-like model has successfully described conventional III–V superlattice structures[59,60] and has recently been applied to MHP materials[61–63]. The dispersion relation for electrons (holes) in the $x$ direction is:

$$\cos\left(\beta L_{qw}\right)\cosh(\alpha L_b) + \frac{1}{2}\left(\gamma - \gamma^{-1}\right)\sin\left(\beta L_{qw}\right)\sinh(\alpha L_b) = \cos(\mathbf{k}(L_{qw} + L_b)) \tag{3}$$

where $L_{qw}$ is the width of the metal halide quantum well layer, and $L_b$ is the width of the barrier layer composed of formamidinium. Both widths are determined from XRD studies ($L_{qw} = 0.624$ nm and $L_b = 0.690$ nm). $\mathbf{k}$ is the superlattice wavevector, which is bound by $-\pi(L_{qw} + L_b)$ and $\pi(L_{qw} + L_b)$. The minimum energy of the lowest-energy miniband occurs when $\mathbf{k} = 0$. For simplicity, we define: $\beta^2 = 2m_{qw,e(h)}E_{e(h)}\hbar^{-2}$ and $\alpha^2 = 2m_{b,e(h)}(V_{e(h)} - E_{e(h)})\hbar^{-2}$. The effective masses of electrons and holes are assumed to be the same for the quantum well ($m_{qw} = m_{qw,e} = m_{qw,h}$) and barrier

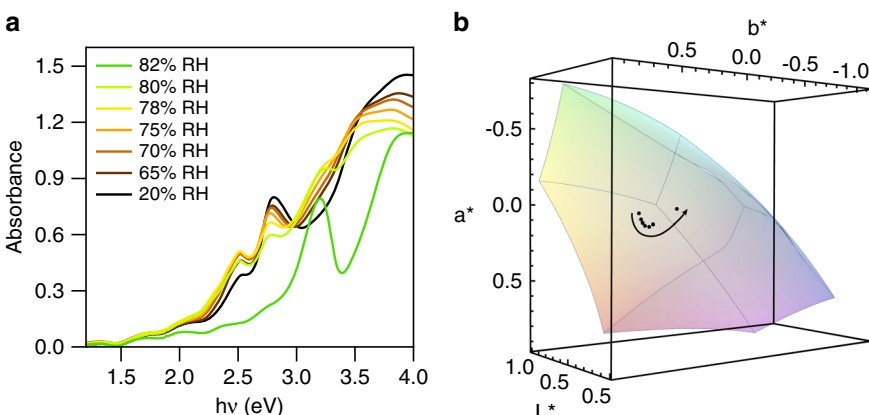

**Fig. 5 Color of hygrochromic films. a** Absorbance and **b** Color of FA$_{n+1}$Pb$_n$I$_{3n+1}$ composite films plotted in CIE 1976 L*a*b* coordinate space. Film color is calculated from absorbance spectra in (**a**). The colored polyhedron in **b** is a visualization of the sRGB gamut used in displays and digital photography. The arrow in **b** shows the direction of increasing relative humidity (RH).

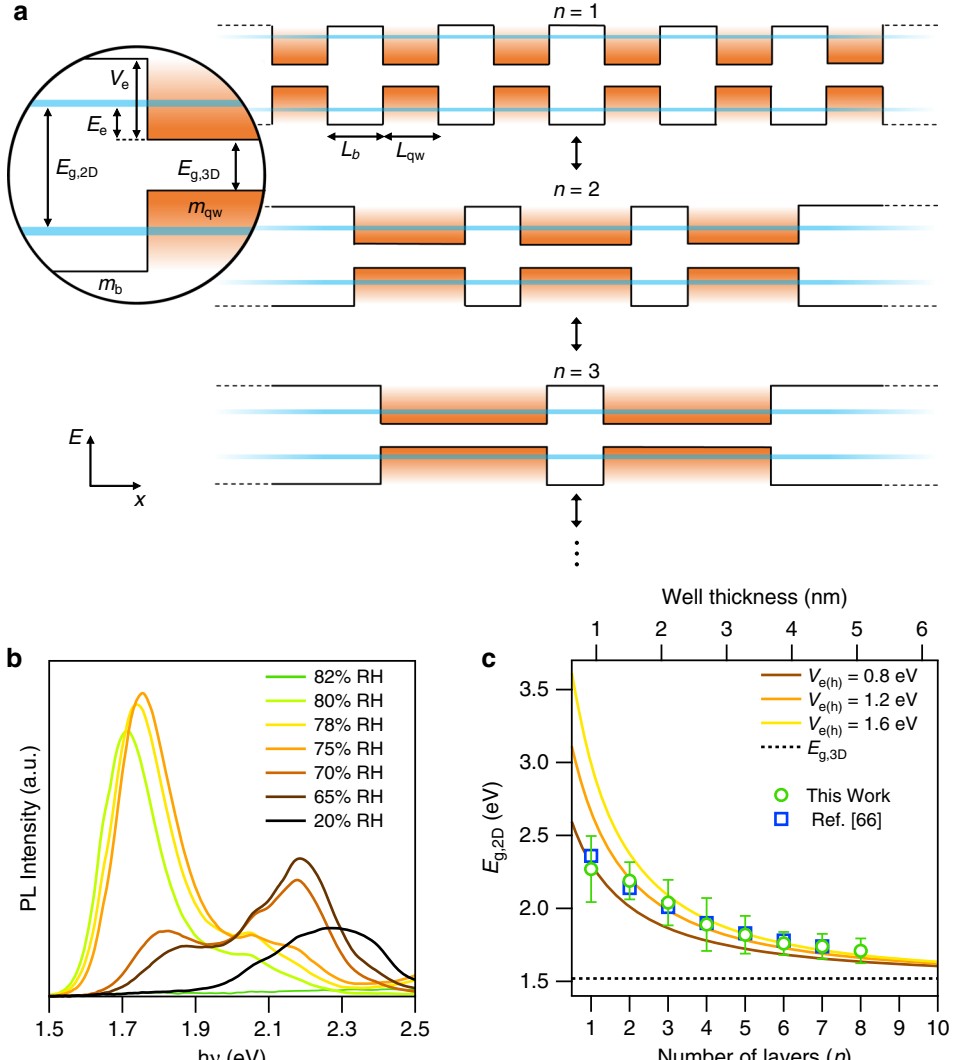

**Fig. 6 Superlattice description of FA$_{n+1}$Pb$_n$I$_{3n+1}$ optical properties. a** Diagram of the Kronig–Penney-like model used to describe FA$_{n+1}$Pb$_n$I$_{3n+1}$ optical properties. **b** Photoluminescence (PL) of hygrochromic FA$_{n+1}$Pb$_n$I$_{3n+1}$ films collected at various relative humidity (RH). Each spectrum exhibits multiple peaks due to a mixture of 'n' layers. **c** A plot of the peak PL position and models used to describe our PL data. Marker position corresponds to the peak of the PL spectra in (**b**), and the error bars are the full-width at half maximum of the peak. Model parameters are described in the text.

($m_b = m_{b,e} = m_{b,h}$). We apply literature values for the effective mass in the metal halide layer[57] ($m_{qw} = 0.1m_0$, where $m_0$ is the rest mass of an electron) and the barrier layer[64] ($m_b = m_{qw}/0.4$). The barrier height ($V_{e(h)}$) for the electrons (holes) is an expression of the bandgap of the formamidinium layers that separate metal halide layers. For simplicity, we assume $V_e = V_h$. The expression for $\gamma$ is modified from the classic KP model ($\gamma = \alpha/\beta$) to take into account the difference in effective mass of the electrons (holes) in the quantum well and barrier layers: $\gamma = \alpha m_{qw,e(h)}/\beta m_{b,e(h)}$.

PL shows discrete miniband transitions from the optical bandgaps of a mixture of FA$_{n+1}$Pb$_n$I$_{3n+1}$ thicknesses ($n$) that increase as the RH increases (Fig. 6b). As the RH increases to 80%, the PL peak shifts from 2.28 eV to 1.71 eV. It is notable that the PL is tuned in the visible region over a 0.56 eV window by simply varying the RH. The PL is quenched upon reaching 82% RH, which is consistent with the transition to δ-FAPbI$_3$[28]. We successfully reproduce our experimental PL by numerically solving Eq. 2 for $E_{e(h)}$ to produce $E_{g,2D}$ as a function of quantum well width ($L_{qw}$) (Fig. 6c). The thickness of a monolayer in FA$_{n+1}$Pb$_n$I$_{3n+1}$ is 0.624 nm. The KP-like model nicely reproduces our optical bandgap data determined from PL measurements for

$n > 2$. A potential barrier height of $V_{e(h)} = 1.2$ eV best fits the data, which is a reasonable bandgap for a FAI salt layer ($E_{g,FAX} = E_{g,3D} + e(V_e + V_h) = 3.92$ eV) between the lead halide sheets. The data is captured by varying the barrier height between 0.8 and 1.6 eV. We posit the barrier height will be affected by the presence of water vapor interacting with the system. The model is also in good agreement with previous work on layered MHP materials, where butyl groups separate methylammonium lead halide layers[43]. The monolayer ($n = 1$) case is not described well by our model, in addition to data from Kanatzitis[43,65] and others[66]. The deviation is known to occur due to the increasing dielectric confinement, which increases the exciton binding energy in $n = 1$ materials[67–69]. Intuitively, the bandgap of a lead iodide monolayer is no longer represented well by the bulk properties of α-FAPbI$_3$.

We expect sizable differences in the photoconductivity of each phase as the FA$_{n+1}$Pb$_n$I$_{3n+1}$ grains transition between 2D FA$_{n+1}$Pb$_n$I$_{3n+1}$, 3D α-FAPbI$_3$, and 1D δ-FAPbI$_3$ based on differences in excitonic character and binding energy. We measure these differences using contactless flash-photolysis time-resolved microwave conductivity (fp-TRMC)[70,71] by photoexciting composite

**Table 1 Average $\phi\Sigma\mu$ values measured by fp-TRMC at 450 nm.**

| RH (%) | Color | n | $\phi\Sigma\mu$ (cm$^2$ V$^{-1}$ s$^{-1}$) |
|---|---|---|---|
| 20 ± 3 | Yellow | 1–2 | 0.003 ± 0.002 |
| 70 ± 3 | Orange | 2–4 | 0.014 ± 0.002 |
| 78 ± 3 | Brown | >4 | 0.019 ± 0.008 |

$FA_{n+1}Pb_nI_{3n+1}$ films at 450 nm and at the lowest possible laser fluences (ca. $10^{11}$–$10^{12}$ photons cm$^{-2}$) to obtain satisfactory signal-to-noise while avoiding higher-order recombination effects associated with high charge densities (Supplementary Fig. 16). Indeed, we observe a significant change in the yield-mobility product ($\phi\Sigma\mu$) when comparing films exposed to 20% RH (yellow) and those to 70–78% RH (orange and brown). The orange and brown phases demonstrate ca. 5–6 times higher $\phi\Sigma\mu$ values compared to the yellow phase (Table 1, Supplementary Table 1), which is likely due in part to increases in charge yield ($\phi$) as the exciton binding energy decreases for more 3D connected (higher n) domains. We believe electron and hole mobilities (represented by $\Sigma\mu$) increase concomitantly with charge yield[72] and argue the $\phi\Sigma\mu$ values of the orange phase are closer to the brown phase likely due to an appreciable density of n > 4 domains that dominate the fp-TRMC signal over smaller (n = 1–3) layers. Putting limits on the sum of the hole and electron mobilities for these compounds, we estimate that values of 0.003–0.3 and 0.019–1.9 cm$^2$ V$^{-1}$ s$^{-1}$ are possible for yellow and brown phases, respectively, assuming charge yields between 100 and 1%.

$\phi\Sigma\mu$ values obtained for the brown (most photoconductive) phase were comparable to those obtained in other works. $(BA)_2(MA)_{n-1}Pb_nI_{3n+1}$ (n = 3–4) was measured to have $\phi\Sigma\mu$ = 0.03–0.04 cm$^2$ V$^{-1}$ s$^{-1}$ [73]. However, these films were measured at up to 2 orders of magnitude higher fluence (ca. $10^{13}$ photons cm$^{-2}$) than the $FA_{n+1}Pb_nI_{3n+1}$ composite films measured here (ca. $10^{11}$ photons cm$^{-2}$). As such, the values for BA-based 2D Ruddlesden–Popper MHPs obtained from that work are likely higher than reported since the low fluence regimes, where $\phi\Sigma\mu$ values exhibit little-to-no fluence-dependence, were not measured. Indeed fp-TRMC measurements on $(BA)_2(MA)_{n-1}Pb_nI_{3n+1}$ (n = 4–5) by Gélvez-Rueda et al.[72] at fluences similar to this work demonstrated $\phi\Sigma\mu$ values ca. 10–30 cm$^2$ V$^{-1}$ s$^{-1}$, over three orders of magnitude larger than the $FA_{n+1}Pb_nI_{3n+1}$ MHPs studied here. Measurements on $(MA)_2Pb(SCN)_2I_2$ (n = 1)[74] and $(BPEA)_2(MA)Pb_2I_7$ (n = 2)[75] yield $\phi\Sigma\mu$ values ca. 1 cm$^2$ V$^{-1}$ s$^{-1}$ which is ca. 50 times greater than $FA_{n+1}Pb_nI_{3n+1}$. A less dramatic difference was found when comparing the range of amplitude-weighted average free charge lifetimes measured for the brown phase (ca. 20–90 ns) to $(MA)_2Pb(SCN)_2I_2$ (n = 1, ca. 100 ns)[74] and $(BA)_2(MA)_{n-1}Pb_nI_{3n+1}$ (n = 4–5, ca. 100–200 ns)[72]. Our scaffold-impregnated films likely have random crystallite orientation, which will lower our $\phi\Sigma\mu$ values relative to some of the literature discussed here[73]. Only free charges within crystallites oriented parallel to our linearly polarized microwave probe will absorb appreciably. The variation of free charge lifetimes (and to some degree $\phi\Sigma\mu$) for the brown phase on a film-to-film basis could be due to differences in the density of crystallites and crystallite size distribution based on modest differences in film processing conditions, such as scaffold thickness and $Al_2O_3$ nanoparticle size dispersity used in the scaffold. We were not able to measure $\phi\Sigma\mu$ values of the white/colorless phase, as the lowest fluences were sufficient to convert this phase into the brown phase via heat-induced dehydration mechanism (Supplementary Fig. 17) that we have observed at temperatures as low as 35 °C (Supplementary Fig. 12).

## Discussion

We synthesized layered FA-based MHPs of the general formula $FA_{n+1}Pb_nX_{3n+1}$ (X = I, Br) and their mixed-halide compositions and show reversible multicolor chromism based on two different stimuli: solvent vapor and temperature. We design composite $FA_{n+1}Pb_nX_{3n+1}$ films by synthesizing $FA_{n+1}Pb_nX_{3n+1}$ domains adjacent to a FAX "reservoir" composed of excess FAX salt that allows FAX salt pairs to shuttle between $FA_{n+1}Pb_nX_{3n+1}$ domains and the FAX reservoir. The interactions between each species results in a structural equilibrium between 2D $FA_{n+1}Pb_nX_{3n+1}$ layers, 3D $\alpha$-FAPbX$_3$ domains, and 1D $\delta$-FAPbI$_3$ domains that can be controlled through modulating the strength and number of H-bonds between the reservoir, MHP, and solvent vapor. Unlike previously reported chromic MHPs that only switch between a single dark and a single transparent phase, these films reversibly switch between multiple colors including yellow, orange, red, brown, and white/colorless. The optical transitions are captured by a Kronig–Penney-like model that describes photoluminescence of the material as metal halide octahedra layers separate and coalesce into superlattices of varied thickness. Our work will enable a new generation of functional materials that couple tunable and reversible chromism with the extraordinary optoelectronic properties of MHP materials.

## Methods

**Materials**. Aluminum oxide nanoparticles ($Al_2O_3$, <50 nm particle size, 20 wt% in isopropanol) and dimethyl sulfoxide (DMSO, anhydrous, ≥99.9%) were purchased from Sigma-Aldrich; lead(II) iodide (PbI$_2$, 99.99%) from TCI; formamidinium iodide (FAI) from GreatCell Solar Materials.

**FA$_2$PbI$_4$ powder preparation**. Powders were prepared by weighing a 4:1 molar ratio of FAI:PbI$_2$ in a nitrogen glovebox, sealing in a hardened stainless-steel vial, then ball-milling in a SPEX high energy ball mill for 60 minutes. The vial was then transferred back to the glovebox for recovery of the milled powders. Powders were not exposed to air/humidity until hygrochromic tests were conducted.

**FA$_{n+1}$Pb$_n$X$_{3n+1}$ composite film preparation**. Glass substrates (25 mm × 25 mm × 1 mm) were sonicated in isopropanol (IPA) for 10 min and blown dry with N$_2$. The substrates were then treated in a UV-ozone cleaner for 10 min before spin-coating 200 μL of 16 wt% $Al_2O_3$ nanoparticles (20–40 nm) in IPA at 3000 rpm for 30 s. The resulting film was annealed at 150 °C for 5 min followed by 500 °C for 25 min. This process yielded a 1.58 ± 0.02 μm-thick $Al_2O_3$ scaffold. The $Al_2O_3$ NP precursor can be diluted with IPA to yield thinner films as shown in Supplementary Fig. 18. Next, 75 μL of a precursor solution containing 3 M FAX and 0.75 M PbX$_2$ (4:1 FAX:PbX$_2$) in DMSO was spin-coated at 4000 rpm for 30 s followed by annealing at 60 °C for 10 min. We note here precursor concentration must be optimized for a given scaffold thickness. Optimal precursor concentrations based on PbX$_2$ ([FAX] = 4[PbX$_2$]) for a given scaffold thickness are as follows: 0.7–1 M PbX$_2$ for 1.0–1.6 μm thick, 0.4–0.6 M PbX$_2$ for 600–1000 nm thick, and 0.3–0.4 M PbX$_2$ for 300–600 nm thick. Both spin-coating and annealing were performed in air while maintaining the relative humidity (RH) between 40 and 45%. The substrates were stored in a drawer (<20% RH) or desiccator in air until needed. See Supplementary Note 2 for a discussion on the impact of processing conditions on composite $FA_{n+1}Pb_nX_{3n+1}$ film formation.

**Humidity control**. Humidy-controlled experiments were performed in a glovebox equipped with a humidity controller connected to a humidity sensor (error of ± 3%), humidifier, and dehumidifier unless noted elsewhere. The controller maintains the humidity with an accuracy of ± 0.1–0.3% The humidity controller was calibrated to saturated KBr in water, which exhibits a relative humidity (RH) of 81.67 ± 0.21% at 20 °C[76]. We reported RH throughout this study; however, RH is different at sea level compared to high altitudes (NREL, Golden, CO, this study). We converted the RH's reported in this study to absolute humidities (AH's) using 25 °C and 630 mmHg as a typical barometric pressure of NREL in Golden, CO (Supplementary Table 2).

**Optical characterization**. Absorbance was collected with a photodiode array Hewlett-Packard 8453 UV-vis spectrometer with $Al_2O_3$ scaffold and glass absorbance subtracted from all spectra. Steady-state PL measurements were taken using a home-built system. Samples were excited using a Thorlabs fiber-coupled 405 nm light-emitting diode (LED) pulsed at 10 Hz using a Thorlabs DC2200 LED driver. Visible detection was made using an Ocean Optics OceanFX spectrometer. Spectra

were stitched using a LabVIEW program developed in-house. Detector calibration was done using an Ocean Optics HL-2000-HP blackbody lamp. Prior to data acquisition, the substrate was sealed into an air-tight optical holder filled with the desired RH. Optical measurements were collected on films prepared with a $326 \pm 22$ nm thick $Al_2O_3$ NP scaffold.

**Structural characterization**. In situ X-ray diffraction data was collected at the Stanford Synchrotron Radiation Light Source (SSRL) at beamline 11-3. The samples were measured at an incident angle of 3 degrees and an incident X-ray wavelength of 0.9744 Å. A Rayonix MX225 2D detector was used to collect data, and a $LaB_6$ standard used to calibrate the data. The RH in the sample chamber was alternated between 35% and 82% RH using a commercial room humidifier powered by a humidity controller, which was connected to a humidity sensor within the sample chamber. The humidity in the sample chamber was reduced by flowing helium gas through the chamber. Once 35% RH was reached, the humidity was increased by manually reducing the He flow until the humidity stabilized at 82% RH. The data collection were continuous except for brief interruptions for changing the sample chamber connection between the humidifier and the He. The integration time per measurement was 30 s. The data was integrated using GSAS-II[77]. $2\theta$ values were calculated by converting Q data relative to Cu Kα (1.5406 Å, 8.04 eV). Scherrer analysis was performed using a κ value of 0.9 and FWHM values with error bars obtained by fitting peaks to a Voigt function. Non-synchrotron WAXS of composite films was collected in air using a Bruker D8 Discover diffractometer with GADDS 4-circle detector (General Area Detector Diffraction System) and Cu Kα (1.5406 Å, 8.04 eV) radiation. WAXS of powder was collected in air using a Rigaku Smartlab diffractometer using Cu Kα (1.5406 Å, 8.04 eV) radiation. Fast scans (ca. 2 min) were done before and after a long scan (ca. 30 min) to verify no change in the material due to beam exposure. Interlayer spacing of butylammonium, phenylethylammonium, and hexylammonium was calculated using Jmol by performing an energy minimization of the molecule, measuring the longest hydrogen–hydrogen distance, and then multiplying this distance by 2 based on the assumption that the interlayer spacing is composed of two end-to-end molecules.

**DRIFTS measurements**. A Bruker Alpha FTIR spectrometer outfitted with a diffuse-reflectance infrared Fourier transform spectroscopy (DRIFTS) attachment was used in the study, and all measurements were performed in a humidity glovebox as described above. Samples were prepared as described above except Au-coated Si was used instead of glass as the substrate. DRIFTS data were collected between 350 and 4000 $cm^{-1}$ with a resolution of 2 $cm^{-1}$ and was atmosphere corrected. RH and temperature data were collected using a data logger with points collected every second. Each DRIFTS spectrum was collected over 45 s, so 45 individual RH and temperature points were averaged to determine the average RH and temperature of each spectrum. The RH was increased at a rate of $3.2 \pm 0.2\%$ $min^{-1}$ and decreased at a rate of $1.9 \pm 0.3\%$ $min^{-1}$ during data acquisition.

**fp-TRMC measurements**. Our flash-photolysis time-resolved microwave conductivity (fp-TRMC) measurement system and methods has been described in detail elsewhere[70,71]. Photoexcitation was accomplished using a Nd:YAG (Spectraphysics Quanta Ray SP Pro 230-30H) laser with 9 W of 355 nm at 30 Hz to pump an OPO (Spectraphysics GWU PremiScan ULD/500) with output over the range of 410–2500 nm with 7 ns pulses (ca. 3 W output, varies by wavelength). Blank quartz substrates and quartz substrates with only the alumina scaffold were used to gauge any background contributions to the fp-TRMC transients and showed no appreciable response. All samples were excited at 450 nm with fluences in the range of ca. $1 \times 10^{11}$ to $2 \times 10^{15}$ photons $cm^{-2}$ depending on the strength of the sample response and phase stability during illumination. A 399 nm long pass filter and 700 nm short pass filter were used to cut out residual 355 and 1064 nm light from the optical parametric oscillator, respectively. Maintaining the RH levels necessary to stabilize the brown phase during measurements was achieved by sealing samples in a leak-resistant microwave cavity within the humidified glovebox described previously. At higher fluences, phases held at higher RH tended to revert back to phases at lower ones, possibly due to heat-driven dehydration of the film (Supplementary Fig. 17). For this reason, the white/colorless phase could not be measured at even the lowest fluences. Phase purity was assessed by monitoring the steady-state photoluminescence for films in-situ during fp-TRMC measurements (e.g., see PL data in Supplementary Fig. 19) using a camera lens to focus emitted photons from the microwave cavity into an optical fiber that was coupled to a Princeton Instruments SpectraPro 2500i spectrometer with a liquid-nitrogen cooled CCD detector. A 500 nm long pass filter was used to filter out excitation light (450 nm) and photoluminescence spectra were typically acquired by averaging 60 seconds to monitor phase stability on a minute-to-minute basis during measurements. PL spectra were not corrected by a calibration lamp. The yellow phase was the only sample for which the entire fluence range was measured. Typical fluences for orange and brown samples were ca. $1–20 \times 10^{11}$ photons/$cm^2$. Most transients were acquired for 30,000 shots or until S/N = 10. For some yellow and orange samples with lower S/N at low fluences, up to 60,000 shots were acquired. The typical fraction of absorbed photons at 450 nm for yellow, orange, and brown phases were 99%, 99%, and 96%, respectively. Transient fitting was done using a

custom global fitting routine in Igor Pro 8 using a biexponential fit. $\phi\Sigma\mu$ values at each fluence were extracted by summing the amplitudes of the optimized fit coefficients. For each sample, an average of $\phi\Sigma\mu$ over the lowest four-to-five fluences (i.e., regime where $\phi\Sigma\mu$ shows little-to-no fluence dependence) were taken to represent the average yield-mobility product for that sample. These values were then further averaged over three samples to obtain the values and uncertainties reported here.

## Data availability
The data that support the findings of this study are available from the corresponding author upon reasonable request.

## Code availability
Mathematica code used to model $FA_{n+1}Pb_nI_{3n+1}$ optical properties is available from the corresponding author upon reasonable request.

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

## Acknowledgements

This study was authored by the National Renewable Energy Laboratory, operated by Alliance for Sustainable Energy, LLC, for the U.S. Department of Energy (DOE) under contract No. DE-AC36-08GO28308. Funding was provided by the Building Technologies Office within the U.S. Department of Energy Office of Energy Efficiency and Renewable Energy. Funding for microwave measurements provided by Department of Energy, Office of Basic Energy Sciences, Division of Chemical Sciences, Biosciences, and Geosciences. Use of the Stanford Synchrotron Radiation Lightsource, SLAC National Accelerator Laboratory, was supported by the U.S. Department of Energy, Office of Basic

Energy Sciences under Contract No. DE-AC02-76SF00515. The views expressed in the article do not necessarily represent the views of the DOE or the U.S. Government. The U.S. Government retains and the publisher, by accepting the article for publication, acknowledges that the U.S. Government retains a nonexclusive, paid-up, irrevocable, worldwide license to publish or reproduce the published form of this study, or allow others to do so, for U.S. Government purposes. The authors thank Vincent M. Wheeler for helpful discussions. T.G.A. and G.R. thank Obadiah G. Reid for helpful discussions on fp-TRMC experiment design and data analysis. The authors thank their respective funding sources for their support.

## Author contributions

B.A.R. and L.M.W. conceived the idea, designed the study, and contributed to all data interpretation. B.A.R. also contributed to all experimental work. L.E.M. and L.T.S. performed in-situ WAXS measurements and analysis. T.G.A. performed fp-TRMC measurements, and T.G.A. and G.R. interpreted the data. D.T.M. performed all ball-milling experiments. K.J.P. and C.A.W. performed and analyzed SEM measurements. B.A.R. and L.M.W. wrote the paper. L.M.W. developed the K-P-like model. All authors discussed the results and revised the paper.

## Competing interests

The authors declare no competing interests.
