## [Peer Review File · Nature Communications]

REVIEWER COMMENTS

Reviewer #1 (Remarks to the Author):

The authors report a layered Ruddlesden-Popper MHP of the general formula FAn+1PbnX3n+1 ($X = \text{I, Br}$; $n = \text{number of layers} = 1, 2, 3, \dots \infty$). Color of the FAn+1PbnX3n+1 domain is controlled by modulating the strength and number of H-bonds in the system by exposure of FAn+1PbnX3n+1 domain to solvent vapor (solvatochromism) or temperature change (thermochromism), which shuttles FAX salt pairs between the layered MHP and an adjacent FAX "reservoir" composed of excess FAX, dimethylsulfoxide, and water. They demonstrate reversible switching from 2-dimensional FA2PbX4 ($n=1$) to 3-dimensional $\alpha\text{-FAPbX3}$ ($n=\infty$) and finally a transition to 1-dimensional $\delta\text{-FAPbI3}$.

1. In the article, the authors described the dynamic equilibrium as equation 1, n represent the number of inorganic slabs, but the letter "x" has not been regarded as a scientific index.

2. We notice that the shift of N-H and H bonding in DRIFTS spectra is tiny at the range from 17.3% to 70.1%. Whether the threshold value of hydrogen bonding is around RH 17%? Whether intermolecularly H-bonded H_2O and H-bonding of solvent has the same equal reversible chromism?

3. Why the $\delta\text{-FAPbI3}$ and $\alpha\text{-FAPbI3}$ has a different Scherrer size?

Reviewer #2 (Remarks to the Author):

The authors synthesized Ruddlesden-Popper type 2D perovskites FAn+1PbnX3n+1 ($X = \text{I, Br}$) and demonstrate reversible transition between 2D phase, 3D and 1D phase. The transition is triggered by solvent or thermal annealing, which led to color change of films. The authors presented details in structural and optical properties during cycling, which are interesting. However, I have a major concern about the structure of perovskite, is it really a 2D perovskite?

1. It has been well know that the FA atom is only 2.53 Å and the tolerance factor of the FA-based perovskite is 0.98, which means that this molecule is almost impossible to separate the inorganic framework, or at least in the film simply fabricated from a solution containing excess FA.

Unfortunately, there is lack of single-crystal information and crystal orientation for these materials. Therefore I can't confirm the truth of the claim of "2D perovskite" for these materials. Meanwhile, the authors refer to the ref.41 to prove its 2D structure. However, as far as I know the claim of 2D structure in this publication is also in doubt.

2. From the absorption spectra (Figure 1a, Figure S4b, Figure S8b), apparently these materials still have absorption until ca. 800 nm. This is more like a 3D FAPbI3 phase rather than 2D analogous.

3. Figure 5 shows excitonic peaks at 2.79 eV, 2.51 eV, 2.30 eV, 2.07 eV, and 1.77 eV, which the authors assigned to $n = 3, 4, \text{ and } 5$, respectively. The authors should give evidence, at least based on calculation for confirmation.

4. Note that during formation of FA-based perovskites, there are many phases are involved including 2H, 6H, and 3C phases. These phases also exhibit different Chromism, which should be carefully examined in the current manuscript.

Reviewer #3 (Remarks to the Author):

Overall this is a comprehensive study on chormism in layered formamidinium metal halide perovskite. Insights into the chormism mechanisms have been given. The manuscript is well written and the science is presented clearly. While this work is of high quality, I think it may be more suitable for a specific journal with a chemistry scope due to the as-demonstrated limited technological/general impact.

Point-by-point response to the reviewers' comments

We thank the reviewers for the overwhelmingly positive reviews. We have performed additional experiments and calculations to directly address these questions and concerns and believe the manuscript is now a more complete and compelling story for publication. The reviewer comments are included below in *italics*. We directly respond to comments in **red**.

Reviewer #1 (Remarks to the Author):

The authors report a layered Ruddlesden-Popper MHP of the general formula $FA_{n+1}Pb_nX_{3n+1}$ ($X = I, Br$; $n = \text{number of layers} = 1, 2, 3, \dots \infty$). Color of the $FA_{n+1}Pb_nX_{3n+1}$ domain is controlled by modulating the strength and number of H-bonds in the system by exposure of $FA_{n+1}Pb_nX_{3n+1}$ domain to solvent vapor (solvatochromism) or temperature change (thermochromism), which shuttles FAX salt pairs between the layered MHP and an adjacent FAX "reservoir" composed of excess FAX, dimethylsulfoxide, and water. They demonstrate reversible switching from 2-dimensional FA_2PbX_4 ($n=1$) to 3-dimensional α - $FAPbX_3$ ($n=\infty$) and finally a transition to 1-dimensional δ - $FAPbI_3$.

1. In the article, the authors described the dynamic equilibrium as equation 1, n represent the number of inorganic slabs, but the letter "x" has not been regarded as a scientific index.

We thank the reviewer for this important point. We have now defined 'x' as a function of 'n', the number of 2D layers. We also decided to change 'x' to 'q' in the new version to distinguish it from the 'x' subscript often used to express single-site (A or X) alloyed compositions in the perovskite literature. Equation 1 has been changed in the main text to the following:

$$n = 1, 2, 3, \dots \infty; q = \frac{1}{n(n+1)}$$

A short derivation is now provided in the supporting information and is reproduced here for the reviewer's convenience:

Derivation of 'q' index.

$$n = 1, 2, 3, \dots \infty; q = \frac{1}{n(n+1)}$$

At a given value of 'n', the amount of FAX relative to Pb can be represented as:

$$\frac{n+1}{n}$$

When we transition to a larger 'n' value, the amount of FAX relative to Pb can be modified to:

$$\frac{(n+1)+1}{n+1}$$

'q' is the amount of FAX gained or lost by transitioning between 'n' values. Therefore,

$$\begin{aligned} q &= \frac{(n+1)+1}{n+1} - \frac{n+1}{n} \\ &= \left(\frac{n}{n}\right) \frac{(n+1)+1}{n+1} - \left(\frac{n+1}{n+1}\right) \frac{n+1}{n} = \frac{n(n+2)}{n(n+1)} - \frac{(n+1)^2}{n(n+1)} = \frac{n(n+2) - (n+1)^2}{n(n+1)} \\ &= \frac{n^2 + 2n - n^2 - 2n - 1}{n(n+1)} = -\frac{1}{n(n+1)} \end{aligned}$$

2. We notice that the shift of N-H and H bonding in DRIFTS spectra is tiny at the range from 17.3% to 70.1%. Whether the threshold value of hydrogen bonding is around RH 17%? Whether intermolecularly H-bonded H₂O and H-bonding of solvent has the same equal reversible chromism?

We thank the reviewer for this observation. Below 17% RH there is a no observable change in the DRIFTS spectra. We believe the reviewer is referring to a threshold at 70.1% RH where chromism is observed. In this case, we find this to be nice consistent insight with our current way of thinking. We restructured the text to first discuss changes in the DRIFTS spectra before the 70% threshold and then a paragraph that discusses DRIFTS after the threshold is reached. The discussion begins with this statement:

"We observe a threshold value for chromism at c.a. 70% RH, which results in more dramatic changes to the DRIFTS spectra. The H-bond interactions shift the thermodynamic equilibrium (Eq. 1) from the FA_{n+1}Pb_nI_{3n+1} domains to the FAI reservoir, indicated by an increase in spectral intensity, peak shifting, and peak broadening at RH >70%."

We also added a panel to Fig. 3b to highlight the observed shifting in the N-H region to better highlight this point.

We are unsure if the reviewer means solvent vapor (H₂O, MeOH, EtOH, IPA) or the solid-state DMSO in the composite film when referring to "H-bonding of solvent", so we address both points. Only solvent vapors that both donate and accept H-bonds, such as water or alcohols (MeOH, EtOH, IPA), will deeply color the chromogenic films. Of these solvent vapors, only H₂O and MeOH will bleach the films, and MeOH requires higher concentrations for longer times for bleaching to occur. We demonstrate this explicitly in Supplementary Fig. 11 and say the following in the main manuscript:

"The ability of the solvent to both donate and accept H-bonds is critical for reversible switching to occur. Solvent vapors such as water or alcohols (methanol, ethanol, and isopropyl alcohol) with H-bond-active hydroxyl groups will deeply color composite FA_{n+1}Pb_nX_{3n+1} films from yellow to orange, red, and brown (Supplementary Fig. 11)."

The role of solid-state DMSO was investigated using $\text{FA}_{n+1}\text{Pb}_n\text{I}_{3n+1}$ that was synthesized by ball-milling 4:1 $\text{FAI}:\text{PbI}_2$ without solvent (DMSO or water). The ball-milled salts exhibit the same WAXS pattern as our composite films. They also exhibit a hydrochromic response, but it is slower and less reversible. We conclude that DMSO is not critical to the shutting of FAX, but may facilitate the process. We added a brief discussion in the main paper:

“Although solvatochromism in our ball-milled powder synthesized without solvent (see Supplementary Fig. 10) shows that DMSO is not crucial to the reversible switching mechanism, DMSO may facilitate switching by providing H-bond acceptor sites.”

3. Why the δ -FAPbI₃ and α -FAPbI₃ has a different Scherrer size?

Within the uncertainty associated with the measurement and fitting procedures, there is no difference in the Scherrer size. The error bars show this explicitly in Fig. 4f. We have added the following discussion to clarify this point:

“... and the size of the domains remain constant as the $\text{FA}_{n+1}\text{Pb}_n\text{I}_{3n+1}$ domains evolve with increasing RH (Fig. 4f).”

Reviewer #2 (Remarks to the Author):

The authors synthesized Ruddlesden-Popper type 2D perovskites $\text{FA}_{n+1}\text{Pb}_n\text{X}_{3n+1}$ ($\text{X} = \text{I}, \text{Br}$) and demonstrate reversible transition between 2D phase, 3D and 1D phase. The transition is triggered by solvent or thermal annealing, which led to color change of films. The authors presented details in structural and optical properties during cycling, which are interesting. However, I have a major concern about the structure of perovskite, is it really a 2D perovskite?

1. It has been well known that the FA atom is only 2.53 Å and the tolerance factor of the FA-based perovskite is 0.98, which means that this molecule is almost impossible to separate the inorganic framework, or at least in the film simply fabricated from a solution containing excess FA. Unfortunately, there is lack of single-crystal information and crystal orientation for these materials. Therefore I can't confirm the truth of the claim of “2D perovskite” for these materials. Meanwhile, the authors refer to the ref.41 to prove its 2D structure. However, as far as I know the claim of 2D structure in this publication is also in doubt.

We have given this comment significant thought and effort. The other reviewers do not share these concerns, but we believe the reviewer illuminates an important point to clarify in the manuscript. Indeed, the FA molecule will fit into the 3D tolerance factor; however, in contrast to the reviewer comments, *that does not preclude it from forming a 2D perovskite*. Oxide and halide-based perovskites are well known to form 2D and 3D crystals using the same atoms. The oxide examples are many. Here are a few: doi.org/10.1038/NMAT4039, doi.org/10.1063/1.5092614, doi.org/10.1007/s40145-020-0365-x. Cs-based lead halide perovskites are excellent halide perovskite examples since Cs is even smaller than FA. In addition to the 3D CsPbX_3 , there are a number of lower dimensionality structures: Cs_4PbX_6 (0D, doi.org/10.1021/acs.chemmater.9b03426), Cs_2PbX_4 (2D, doi.org/10.1039/C9TC02267H), and CsPb_2X_5 (2D, doi.org/10.1002/cssc.201701131). There are other reports in literature on the synthesis and characterization of analogous A_2PbX_4 compounds cited as ref. 38 (formerly 41) and ref. 39 (formerly 42). While we cannot speak to the results of ref. 38 (formerly 41), we cite this

reference along with others solely as a comparison to other existing A_2PbX_4 analogues. Specifically, we say the following after an in-depth discussion on the identity of phases in our WAXS pattern:

“... and our WAXS results are in excellent agreement with previously reported MA_2PbI_4 ³⁸ and FA_2PbBr_4 .³⁹”

We further respond to this comment with additional experiments and modeling. We performed experiments to produce $FA_{n+1}Pb_nI_{3n+1}$ using a *second* synthesis method. $FA_{n+1}Pb_nI_{3n+1}$ was synthesized by simply ball-milling 4:1 FAI:PbI₂ without solvent (DMSO or water) in an inert atmosphere. All extraneous materials (solvent, air) were removed. WAXS of ball-milled $FA_{n+1}Pb_nI_{3n+1}$ perfectly matches our thin films and are now presented in a new Supplementary Fig. 1 (reproduced below). It conclusively shows the WAXS peak located at 0.703 \AA^{-1} corresponds to FA_2PbX_4 ($n=1$), and not the commonly reported Pb(DMSO) complex, which also has a strong peak in this region, or any other phase of $FAPbX_3$ (see response to reviewer 2 comment 4). There are three peaks in addition to the 0.703 \AA^{-1} peak that match FA_2PbI_4 ($n=1$) and no other compounds. Our new Supplementary Fig. 1 (reproduced below) summarizes our ball-milling analysis and compares the new ball-milling results with our previous film results:

Supplementary Figure 1 | $FA_{n+1}Pb_nX_{3n+1}$ powder prepared by ball-milling 4:1 FAX:PbX₂. WAXS comparison of $FA_{n+1}Pb_nI_{3n+1}$ powder and films.

We modeled the WAXS pattern of FA_2PbX_4 ($n=1$) using the following reference (doi.org/10.1063/1.4947305) to provide further clarity that we are indeed synthesizing 2D perovskites. All WAXS peaks unaccounted for by α -FAPbX₃ match the included standard patterns for staggered (instead of eclipsed) FA_2PbX_4 . We modified the language of our manuscript to reflect this:

“WAXS of composite $FA_{n+1}Pb_nX_{3n+1}$ films (Fig. 1b) match $FA_{n+1}Pb_nI_{3n+1}$ powder produced by ball-milling (Supplementary Fig. 1) and both exhibit a mixture of Bragg diffraction peaks that correspond to 2D FA_2PbX_4 ($n=1$) with staggered octahedral layers and 3D α -FAPbX₃ (3C/3R). Specifically, we observe Bragg diffraction peaks at 0.703 \AA^{-1} , 1.400 \AA^{-1} , 2.040 \AA^{-1} and 2.095 \AA^{-1}

for 100% I and 0.738 \AA^{-1} , 1.478 \AA^{-1} , 2.157 \AA^{-1} , and 2.198 \AA^{-1} for 100% Br that correspond to the (001), (002), (201), and (003) planes of 2D FA_2PbX_4 ($n=1$) (Fig. 1b, asterisks).”

We also modified our illustration in Fig. 2 to show the staggered perovskite planes.

2. From the absorption spectra (Figure 1a, Figure S4b, Figure S8b), apparently these materials still have absorption until ca. 800 nm. This is more like a 3D FAPbI₃ phase rather than 2D analogous.

The reviewer is correct. We also believe the absorption at ca. 800 nm is due to 3D α -FAPbI₃. The films are a mixture of many phases that span 2D layered materials with varying ‘n’ up to bulk-like 3D α -FAPbI₃. We are unable to make phase-pure films, but we demonstrate control over the relative concentrations of the different phases. This notion was described extensively in the WAXS discussion (see response to reviewer 2, comment 1), and we have added clarification in the discussion around the absorption spectra:

“Our optical absorption and photoluminescence (PL) data confirms each observed color is a mixture $\text{FA}_{n+1}\text{Pb}_n\text{I}_{3n+1}$ with multiple thicknesses that span $n=1$ to $n=\infty$ (Fig. 5a).”

3. Figure 5 shows excitonic peaks at 2.79 eV, 2.51 eV, 2.30 eV, 2.07 eV, and 1.77 eV, which the authors assigned to $n = 3, 4,$ and $5,$ respectively. The authors should give evidence, at least based on calculation for confirmation.

We thank the reviewer for pointing this out. This comment has led to the most extensive changes to strengthen our manuscript. We developed a Kronig-Penney-like model that nicely reproduces our data and similar data in the literature by modeling the material as a superlattice structure formed by alternating organic and inorganic sheets. Our new Fig. 6 summarizes our findings:

Fig. 6 | Superlattice description of $FA_{n+1}Pb_{n+1}I_{3n+1}$ optical properties. (a) Diagram of the Kronig-Penney-like model used to describe $FA_{n+1}Pb_{n+1}I_{3n+1}$ optical properties (b) PL of hygrochromic $FA_{n+1}Pb_{n+1}I_{3n+1}$ films collected at various RH. Each spectrum exhibits multiple peaks due to a mixture of ‘n’ layers. (c) A plot of the peak PL position and models used to describe our PL data. Marker position corresponds to the peak of the PL spectra in (b), and the error bars are the full-width at half maximum of the peak. Model parameters are described in the text.

We now include the following discussion around these results:

“Discrete optical transitions observed in the absorbance spectra occur due to the separation or coalescence of 2D octahedra layers. The optical bandgap of 2D $FA_{n+1}Pb_{n+1}I_{3n+1}$ materials increases monotonically as n approaches 1 due to formation of minibands in the

quantum well superlattice structure that emerges due to alternating layers of formamidinium and connected lead halide octahedra layers (Fig. 6a). The increase from 3D α -FAPbI₃ is written as:

$$E_{g,2D} = E_{g,3D} + E_e + E_h \quad (2)$$

where $E_{g,3D} = 1.52$ eV is the bulk bandgap of α -FAPbI₃⁵⁸ and $E_{e(h)}$ is the minimum energy of the lowest-energy miniband. We determine $E_{e(h)}$ by adapting the Kronig-Penney (KP) model⁵⁹ for an electron (hole) in a one-dimensional periodic potential. The KP-like model has successfully described conventional III-V superlattice structures^{60,61} and has recently been applied to MHP materials.⁶²⁻⁶⁴ The dispersion relation for electrons (holes) in the x direction is:

$$\cos(\beta L_{qW}) \cosh(\alpha L_b) + \frac{1}{2}(\gamma - \gamma^{-1}) \sin(\beta L_{qW}) \sinh(\alpha L_b) = \cos(k(L_{qW} + L_b)) \quad (3)$$

where L_{qW} is the width of the metal halide quantum well layer, and L_b is the width of the barrier layer composed of formamidinium. Both widths are determined from XRD studies ($L_{qW} = 0.624$ nm and $L_b = 0.690$ nm). k is the superlattice wavevector, which is bound by $-\pi(L_{qW} + L_b)$ and $\pi(L_{qW} + L_b)$. The minimum energy of the lowest-energy miniband occurs when $k = 0$. For simplicity, we define: $\beta^2 = 2m_{qW,e(h)}E_{e(h)}\hbar^{-2}$ and $\alpha^2 = 2m_{b,e(h)}(V_{e(h)} - E_{e(h)})\hbar^{-2}$. The effective masses of electrons and holes are assumed to be the same for the quantum well ($m_{qW} = m_{qW,e} = m_{qW,h}$) and barrier ($m_b = m_{b,e} = m_{b,h}$). We apply literature values for the effective mass in the metal halide layer⁵⁸ ($m_{qW} = 0.1m_0$, where m_0 is the rest mass of an electron) and the barrier layer⁶⁵ ($m_b = m_{qW}/0.4$). The barrier height ($V_{e(h)}$) for the electrons (holes) is an expression of the bandgap of the formamidinium layers that separate metal halide layers. For simplicity, we assume $V_e = V_h$. The expression for γ is modified from the classic KP model ($\gamma = \alpha/\beta$) to take into account the difference in effective mass of the electrons (holes) in the quantum well and barrier layers: $\gamma = \alpha m_{qW,e(h)}/\beta m_{b,e(h)}$.

PL shows discrete miniband transitions from the optical bandgaps of a mixture of FA_{*n*+1}Pb_{*n*}I_{3*n*+1} thicknesses (n) that increase as the RH increases (Fig. 6b). As the RH increases to 80%, the PL peak shifts from 2.28 eV to 1.71 eV. It is notable that the PL is tuned in the visible region over a 0.56 eV window by simply varying the RH. The PL is quenched upon reaching 82% RH, which is consistent with the transition to δ -FAPbI₃.²⁶ We successfully reproduce our experimental PL by numerically solving Equation 2 for $E_{e(h)}$ to produce $E_{g,2D}$ as a function of quantum well width (L_{qW}) (Fig. 6c). The thickness of a monolayer in FA_{*n*+1}Pb_{*n*}I_{3*n*+1} is 0.624 nm. The KP-like model nicely reproduces our optical bandgap data determined from PL measurements for $n > 2$. A potential barrier height of $V_{e(h)} = 1.2$ eV best fits the data, which is a reasonable bandgap for a FAI salt layer ($E_{g,FAI} = E_{g,3D} + e(V_e + V_h) = 3.92$ eV) between the lead halide sheets. The data is captured by varying the barrier height between 0.8 and 1.6 eV. We posit the barrier height will be affected by the presence of water vapor interacting with the system. The model is also in good agreement with previous work on layered MHP materials, where butyl groups separate methylammonium lead halide layers.³⁷ The monolayer ($n = 1$) case is not described well by our model, in addition to data from Kanatzitis^{37,66} and others.⁶⁷ The deviation is known to occur due to the increasing dielectric confinement, which increases the exciton binding energy in $n = 1$ materials.⁶⁸⁻⁷⁰ Intuitively, the bandgap of a lead iodide monolayer is no longer represented well by the bulk properties of α -FAPbI₃."

4. Note that during formation of FA-based perovskites, there are many phases are involved including 2H, 6H, and 3C phases. These phases also exhibit different Chromism, which should be carefully examined in the current manuscript.

The reviewer brings up an important point. We have added additional analysis on the different phases of FA-based perovskites including 2H, 4H, 6H, 3C, and 3R. We added a new Supplementary Fig. 2 (reproduced below), and a discussion to the main text about the different phases, and how our analysis leads us to conclude we are forming layered compounds. Specifically, we say the following:

“Several different structures have been observed during formation of FA-based halide perovskites including 2H, 4H, 6H, 3C, and 3R structures (see Supplementary Fig. 2).⁴⁰ Our WAXS data eliminates any significant contributions from hexagonal structures (2H, 4H, 6H).”

Supplementary Figure 2 | Comparison of FA_{n+1}Pb_nI_{3n+1} film WAXS data to structures commonly observed during formation of FA-based halide perovskites. Standard patterns were obtained from ref. [5] and modified to include only FA⁺, Pb²⁺, and I⁻ ions by multiplying the unit cell volume by the appropriate ionic radius ratio. We note that 2H corresponds to δ-FAPbI₃, 3C corresponds to single-crystalline α-FAPbI₃, and 3R corresponds to thin-film α-FAPbI₃.

Reviewer #3 (Remarks to the Author):

Overall this is a comprehensive study on chromism in layered formamidinium metal halide

perovskite. Insights into the chromism mechanisms have been given. The manuscript is well written and the science is presented clearly. While this work is of high quality, I think it may be more suitable for a specific journal with a chemistry scope due to the as-demonstrated limited technological/general impact.

We thank the reviewer for their comments. Though we do not demonstrate a particular technology, we believe our work is enabling for many future technologies that leverage semiconductors with dynamic, rather than static, properties. Intuitive applications span next-generation switchable photovoltaics, neuromorphic computing, and battery electrodes. We have added the following sentence to the introduction:

“Realizing reversible chromism in MHPs unlocks a new class of functional materials that couples a dynamic element to their remarkable optoelectronic properties. We envision dynamically tunable semiconductors to have applications that span switchable photovoltaics^{21,22} to energy storage²³ and neuromorphic computing.²⁴”

REVIEWERS' COMMENTS:

Reviewer #1 (Remarks to the Author):

The author has addressed all comments. It can be accepted as is.

Reviewer #2 (Remarks to the Author):

I am satisfy with the improvements. The manuscript looks much better now. I suggest publication of the paper.